# Genome-Wide Association Study Reveals the Genetic Basis of Duck Plumage Colors

**DOI:** 10.3390/genes14040856

**Published:** 2023-03-31

**Authors:** Xinye Zhang, Tao Zhu, Liang Wang, Xueze Lv, Weifang Yang, Changqing Qu, Haiying Li, Huie Wang, Zhonghua Ning, Lujiang Qu

**Affiliations:** 1National Engineering Laboratory for Animal Breeding, Department of Animal Genetics and Breeding, College of Animal Science and Technology, China Agricultural University, Yuanmingyuan West Road 2, Beijing 100193, China; xinye_leaf@163.com (X.Z.); zhutao@cau.edu.cn (T.Z.); ningzhh@cau.edu.cn (Z.N.); 2Beijing Municipal General Station of Animal Science, Beijing 100107, China; wangliangcau@139.com (L.W.); lvxueze0310@163.com (X.L.); carspstp@126.com (W.Y.); 3Engineering Technology Research Center of Anti-Aging Chinese Herbal Medicine of Anhui Province, Fuyang Normal University, Fuyang 236037, China; qucq518@163.com; 4College of Animal Science, Xinjiang Agricultural University, Urumchi 830052, China; lhy-3@163.com; 5College of Animal Science, Tarim University, Alar 843300, China; whedky@126.com

**Keywords:** duck plumage color, *MC1R* gene, *MITF* gene, genome-wide association study, epitasis

## Abstract

Plumage color is an artificially and naturally selected trait in domestic ducks. Black, white, and spotty are the main feather colors in domestic ducks. Previous studies have shown that black plumage color is caused by *MC1R*, and white plumage color is caused by *MITF*. We performed a genome-wide association study (GWAS) to identify candidate genes associated with white, black, and spotty plumage in ducks. Two non-synonymous SNPs in *MC1R* (c.52G>A and c.376G>A) were significantly related to duck black plumage, and three SNPs in *MITF* (chr13:15411658A>G, chr13:15412570T>C and chr13:15412592C>G) were associated with white plumage. Additionally, we also identified the epistatic interactions between causing loci. Some ducks with white plumage carry the c.52G>A and c.376G>A in *MC1R*, which also compensated for black and spotty plumage color phenotypes, suggesting that *MC1R* and *MITF* have an epistatic effect. The *MITF* locus was supposed to be an upstream gene to *MC1R* underlying the white, black, and spotty colors. Although the specific mechanism remains to be further clarified, these findings support the importance of epistasis in plumage color variation in ducks.

## 1. Introduction

Plumage color in birds plays a significant role in courtship and camouflage [1,2]. It is also an important target trait in the poultry industry and has been used for sex identification and breed improvement [3]. In ducks, a greater variety of plumage colors have been found in domestic ducks compared to the mallard [4] (Figure 1). White plumage has been found in the Beijing duck, Lian-cheng white duck, etc.; black plumage has been found in the Wendeng black duck and Putian black duck, etc.; and spotty plumage has been found in the Gaoyou duck and Shanma duck, etc. These plumage variations are used to clearly distinguish domesticated ducks from their wild ancestors, which indicates that alleles for black and white plumage color are undergoing artificial selection during the domestication process of ducks.

There are numerous studies focusing on the molecular mechanisms of plumage color variation. Studies have found that the variation in the content, proportion, and location of melanin in plumage affects different plumage color phenotypes [5]. Melanin biosynthesis takes place in melanosomes [6,7,8]. Melanin mainly consists of two types, eumelanin and pheomelanin [9]. The two types of melanin are both tyrosine derivatives. The relative amounts of the two melanin pigments are determined by tyrosinase activity and expression, with high tyrosinase activity resulting in the production of eumelanin and low tyrosinase activity resulting in the production of pheomelanin [10]. The migration, proliferation, and differentiation of melanocytes can also affect pigmentation, leading to differences in plumage color [11].

Previous studies have identified 11 loci associated with plumage color in ducks. Among these, at the recessive white (c) locus, the recessive allele (cc) is responsible for the white plumage color [4,12,13]. Gong et al. [14] found the recessive allele (tt) at the autosomal locus (T) determines the white plumage in *Liancheng* duck. The E locus is related to black plumage. In chickens, the black locus (E) corresponds to *MC1R* [15,16]. However, the relationship between loci and regulatory genes that affect pigmentation traits in ducks is still unclear.

Currently, numerous genes related to melanin have been identified. *MITF* belongs to the family of the bHLH-leucine zipper transcription factors and is involved in the melanogenesis pathway [17]. *MITF* regulates a variety of melanocyte-specific genes, such as directly regulating the genes encoding *TYR*, *TYRP1*, and *TYRP2/DCT*, resulting in an upregulation of their expression. Mutations in the *MITF* gene can cause different depth of coat color depigmentation in mice, including dilutions, white spotting, or a complete absence of pigmentation [18,19,20]. Silver plumage in Japanese quail is linked to the B mutation in the *MITF* gene [21]. *MITF* is also a key gene associated with the white plumage in ducks [22,23,24,25,26,27]. *MC1R* can promote the production of eumelanin by stimulating the intracellular cAMP signal transduction pathway [28]. If MC1R at low activity, pigment synthesis will generally shift to a predominantly pheomelanin. In chickens, *MC1R* encodes the E locus and affects the plumage color phenotype. Different alleles of E locus affect the distribution of eumelanin and phaeomelanin, leading to different depth of pigment phenotypes in chicken feathers [29]. Glu92Lys mutations in *MC1R* cause black plumage in chickens [15,16,30]. Zhang et al. [31] found that three mutations in *MC1R* (one synonymous SNP (69 C>T) and two non-synonymous SNPs (212 T>C and 274 A>G)) were related to tile-grey plumage in chickens. Zhang et al. [32] found the Ile57Val mutation was significantly related with the black color phenotype in Japanese quails. In ducks, the extended black color is also associated with *MC1R* [33,34].

Plumage color is affected by many complex genetic factors, involving linkage, epistasis, and pleiotropy [35]. Gene interactions also play an important role in plumage color. However, epistatic interaction effects between *MC1R* and *MITF* on duck coloration have not been determined. Epistasis is generally defined as non-additive effects of two genes, whereby the effect of one locus is masked by that of another locus [36]. Analyses of epistasis can provide a deeper understanding of the genetic basis of complex traits [36,37,38]. Rapid advances in bioinformation technology have enabled the identification of candidate variants or genes at the whole-genome scale, as well as gene interactions [37,39,40].

In this study, we explored the genetic mechanism underlying duck plumage colors. In particular, we selected 96 ducks with different colors for next-generation sequencing and GWAS of white, black, and spotty colors to identify candidate genes. Our study focused on the roles of gene interactions in plumage color and on providing a new insight for genetic research on the plumage color in ducks.

## 2. Materials and Methods

### 2.1. Sampling

In total, 96 ducks from two wild populations and ten domestic breeds were used, including wild spot-billed duck (SB, *n* = 7), mallard (MD, *n* = 8), Peking duck (PK, *n* = 8), Maple Leaf duck (FY, *n* = 8), Cherry Valley Duck (CV, *n* = 8), Wendeng black duck (WD, *n* = 8), Putian black duck (PT, *n* = 8), Gaoyou duck (GY, *n* = 8), Jinding duck (JD, *n* = 8), Mei duck (MEI, *n* = 9), Shanma duck (SM, *n* = 8), Shaoxing duck (SX, *n* = 8). SB was sampled from Ningxia Province, China; MD was sampled from Zhejiang Province, China; PK, FY, and CV were sampled from Beijing, China; JD, SM, and PT were sampled from Fujian Province, China; SX and GY ducks were sampled from Jiangsu Province, China; MEI was sampled from Anhui Province, China; WD was sampled from Shandong, China. Detailed sample information, including plumage colors, is provided in Table 1. The animal experimentation protocols were approved by the Animal Welfare Committee of China Agricultural University, and the study was performed in accordance with ethical guidelines for animal research.

### 2.2. Genotyping and SNP Calling

Blood samples were collected through the wing vein. The genomic DNA was extracted with a standard phenol-chloroform method. The DNA samples were sequenced on the Illumina Hiseq2500 platform, with the read length of 150 bp at each end (San Diego, CA, USA). The sequences were filtered with default parameters using fastp v0.20.0 [41]. Reads with Q value less than 20 or high-quality bases less than 70% were excluded.

The clean reads were mapped to the duck reference genome (cau_duck1.0) using BWA-MEM v0.7.15 [42]. Genome Analysis Toolkit version 4.1.8.1 was used for the realignment of reads and SNP calling. The ‘SortSam’ command was used for reads sorting by coordinate. Duplicate reads were marked using the ‘MarkDuplicates’ command. The ‘ApplyBQSR’ command was used to recalibrate the base quality score. The ‘HaplotypeCaller’ command was used to call SNPs and indels, and ‘CombineGVCFs’ command was used to generate variant set in VCF format. A joint genotyping was performed with the ‘GenotypeGVCFs’ command.

Then, the SNPs were filtered by following thresholds: (a) QUAL > 30.0; (b) QD > 5.0; (c) FS < 60.0; (d) MQ > 40.0; (e) MQRankSum > 12.5; and (f) ReadPosRankSum > 8.0. Additionally, 10 bp windows with more than three SNPs were excluded from analyses [43].

### 2.3. GWAS

Plink was used for quality filtering prior to GWAS. After filtration, we obtained a total of 10,014,685 SNPs. Subsequently, GEMMA [44] was used for the GWAS with a univariate linear mixed model. The models were as follows:y = W*α* + x*β* + u + *ϵ*,(1)
where y denotes the vector of phenotypic values for individuals, W is the matrix of covariates, *α* denotes the vector of corresponding coefficients consisting of the intercept, x is the vector of marker genotypes, *β* is an estimate of the additive effect of the marker/SNP, u is the vector of random effects, *ϵ* is the vector of errors.

The significance threshold was set to 0.05/SNP number. The GWAS results were visualized using the R package CMplot. The genome control inflation factor lambda (*λ*) was calculated to evaluate confounding due to population stratification using the R package gap.

### 2.4. Analysis of SNP–SNP Epistasis

Epistatic effects were detected using PLINK v1.9 with command ‘–fast-epistasis’. The statistical model used to test for epistatic effects associated with plumage color traits was as follows:$$\ln\left(\frac{P\left(Y=\mathrm{case}\right)}{P\left(Y=\mathrm{control}\right)}\right){=\beta }_{0}{+\beta }_{1}g_{A}{+\beta }_{2}g_{B}{+\beta }_{3}g_{A}g_{B}$$
where, for each inspected variant pair (A, B), *g_A_* and *g_B_* were the allele counts. The *β*_3_ coefficients were tested for significance. We identified epistasis between SNPs associated with genes using *p*-value threshold of 0.05.

### 2.5. Functional Annotation

Candidate genes were identified by the overlapping significant SNPs on the reference genome (cau_duck1.0), and the Variation Effect Predictor tool in Ensembl was used to evaluate the functional enrichment of candidate SNPs (http://useast.ensembl.org/Multi/Tools/VEP, accessed on 11 February 2023) [45].

## 3. Results

### 3.1. Associations between SNPs and Plumage Color

Since the λ value of the different plumage color were 1.029 (black plumage) and 0.987 (white plumage), the effect of population stratification was not considered in our study. By a GWAS, we identified 64 SNPs using the threshold of significant *p*-value (4.583099 × 10^−9^) that were significantly associated with the black/spotty plumage color. Ten genes were around significant peaks, including *MC1R*, *TCF25*, *NELL2*, *CARD10*, *PTGFRN*, *AKIRIN2*, *RYR3*, *PGAP1*, *FRMPD3*, *USP20*. Among them, *MC1R* is associated with pigmentation [16,30,46,47]. The GWAS results and SNP information are shown in Figure 2 and Appendix A.

For the white/spotty plumage color group, we identified 491 SNPs using the threshold of significant *p*-value (6.042978 × 10^−9^). Thirty-two genes were around significant peaks, including *WNT7B*, *VWF*, *MTUS2*, *CACNB2*, *PKHD1*, *CEP162*, *PDGFC*, *SLIT2*, *GRIA2*, *DLST*, *HERC4*, *SEMA5B*, *ZNF385B*, *SLC49A4*, *ACVR2A*, *LPP*, *PCSK6*, *ADCY7*, *WTIP*, *GPI*, *MGLL*, *MAGI1*, *ARL8B*, *ITPR1*, *PPP4R2*, *MITF*, *SLC25A26*, *PTPRG*, *ITIH4*, *QRICH1*, *ARHGAP26*, *MVB12B*. Some genes, such as *WNT7B* and *MITF*, have been demonstrated to be associated with pigmentation [17,48]. Among these genes, *MITF* is known to be essential for melanocyte development and could directly account for the phenotypic difference between white plumage and spotty plumage in this study. The significant SNPs in *WNT7B* have a weaker relation to the white plumage. The detail GWAS results and SNPs are shown in Figure 3 and Appendix A.

Polymorphisms in *MC1R* and *MITF* are listed in Table 2. We detected two nonsynonymous SNPs (c.52G>A and c.376G>A) in *MC1R* and found that the GA and AA genotypes of c.52G>A and c.376G>A in *MC1R* showed higher frequencies in WD and PT, suggesting that the two mutations in *MC1R* was significantly associated with black plumage. At the c.52G>A locus, AA was the most frequent genotype in PT with 62.5% while GA was the most frequent genotype in WD with 75%. At the c.376G>A locus, AA was the most frequent genotype in PT with 62.5% while GA was the most frequent genotype in WD with 75%. In WD, two individuals carry the GG genotype in c.52G>A, and one carries the GG genotype in c.376G>A. Furthermore, we observed that genotype GA and AA genotypes of SNPs (c.52G>A and c.376G>A) showed higher frequencies in PK, CV, and FY, as opposed to their white plumage phenotype, suggesting that *MC1R* and *MITF* have an interaction effect. Three SNPs (chr13:15411658A>G, chr13:15412570T>C, and chr13:15412592C>G) were detected in *MITF*, which is located in the intron regions. The AA genotype of chr13:15411658A>G, the TT genotype of chr13:15412570T>C, and the CC genotype of chr13:15412592C>G were the most frequent genotype in PK, FY, and CV, which were responsible for the recessive white variant in ducks.

In addition, to detect the 6.6-kb insertion in Pekin ducks found by Zhou et al. [27], we examined the deletion of a large segment of *MITF* in spotty-feathered ducks and found this insertion in all spotty ducks is missing.

### 3.2. MITF–MC1R Interaction and Epistasis

We examined the interaction between *MITF* and *MC1R* and epistatic effects by analyzing SNPs in candidate genes. In an epistasis analysis, a significant interactive effect of *MC1R* and *MITF* was observed in the white and spotty plumage group (*p* < 0.05) and white and black plumage group (*p* < 0.05) but not in white and non-white plumage group (*p* > 0.05) (Table 3).

## 4. Discussion

Plumage color is an important trait in the poultry industry. The genetic and molecular mechanisms underlying the diversity in plumage color have been studied extensively. *MC1R* and *MITF* have been identified as candidate genes related to this trait [26,27,33,34]. The *MC1R* gene encodes the *MC1R* protein, a G protein-coupled receptor primarily expressed in melanocytes. The activation of *MC1R* is mediated by α-melanocyte-stimulating hormone (α-MSH) and leads to an increase in eumelanin. Conversely, binding to the inverse agonist Agouti inhibits *MC1R* activity, which reduces eumelanin and increases pheomelanin within melanocytes [28]. *MITF* has been identified as the main regulatory factor in melanocyte biology and can activate the expression of genes involved in the production of melanin [49]. In this study, we evaluated the mutations of *MC1R* and *MITF* involved in plumage color variation in ducks. Two non-synonymous SNPs in *MC1R* (c.52G>A and c.376G>A) were associated with black plumage in ducks in this study. The AA genotype of c.52G>A and the AA genotype of c.376G>A showed higher frequencies in PT. The GA genotype of c.52G>A and the GA genotype of c.376G>A showed higher frequencies in WD. As previously reported, the extended black phenotype in ducks is caused by non-synonymous SNPs (c.52G>A and c.376G>A) in *MC1R* [33,34]. p.Glu18Lys, resulting from the c.52G>A substitution, may have led to the overexpression of *MC1R* by either disrupting or creating a salt bridge with the extracellular ligands of the *MC1R* protein [33]. Furthermore, p.Val126Ile, resulting from the c.376G>A mutation in the third transmembrane domain of *MC1R*, has been linked to pigmentation in both chickens and ducks [16,33,50].

SNPs, indels, and structural variants in *MITF* have been reported as possible causes of duck white plumage. Li et al. [22] found that two isoforms of *MITF* (B and M isoforms) were expressed in black feather bulbs while only an M isoform of *MITF* was expressed in white feather bulbs. Lin et al. [23] found that g.39807T>G and g.40862G>A of *MITF* were associated with white plumage in Putian ducks while g.32813G>A of *MITF* was strongly correlated with white plumage in Liancheng white ducks. Zhang et al. [26] found thirteen SNPs and two indels associated with white plumage in ducks. Yang et al. [25] have found that six SNPs of *MITF* are associated with plumage color in Kaiya–Liancheng F2 offspring. Guo et al. [51] found that *MITF* is closely related to white plumage color. In this study, we identified three SNPs in *MITF* (chr13:15411658*A*>*G*, chr13:15412570*T*>*C*, and chr13:15412592*C*>*G*) in white-feathered ducks, which are located in the intron region. The above findings showed that *MITF* was significantly associated with white plumage color. Additionally, small indels and large intron insertions are complex variants with the potential to have substantial impacts on phenotypic variation [52,53,54]. In our study, we detected the 6.6kb insertion in ducks and found this insertion was present in ducks with white plumage. This suggests that functional mutations in *MITF* are key factors in the white plumage phenotype of ducks.

Epistasis plays a crucial role in the evolution of complex genetic systems. Interactions between genes can have important implications for the functions of genetic pathways [55]. Our results showed that *MITF* and *MC1R* have an epistatic relationship. In the white/spotty plumage groups and white/black plumage groups, the white feather gene (*MITF*) masked the effect of the extended black feather gene (*MC1R*). In the white/non-white plumage groups, we could not detect an epistatic effect between *MITF* and *MC1R*. Research has shown that there is a complex interaction between *MITF* and *MC1R* in the regulation of melanocyte function and melanin synthesis. *MITF* regulates the expression of enzymes responsible for melanin synthesis and the expression of a receptor that is involved in the function of melanocytes [56]. In cream-colored Australian cattle dogs, researchers have found that a single nucleotide variant in the *MC1R* promoter causes cream coloration in dogs [57]. Zhou et al. [27] found that the expression levels of genes downstream of *MITF* in the melanin synthesis pathway (*MLANA*, *TYR*, *TYRP1*, *DCT*, *OCA2*, and *MLPH*) were downregulated, indicating that *MITF* plays a switching role in the melanogenesis pathway of Pekin duck. However, the molecular mechanism remains to be further clarified.

Studies of duck plumage color [4,58] have revealed that at the extended black locus (E), the E allele causes black plumage while the non-extended black homozygous ee genotype results in a spotty plumage color. Lin et al. [59] found that the average frequency of spotty plumage in offspring produced by Putian black ducks was 23.15% while the remaining offspring show black plumage, indicating that black plumage was dominant. Homozygous recessive alleles (cc) at the C locus produce white plumage, which prevents the expression of other colors [4,13,58]. This suggests that, regardless of the genotype at the E locus, individuals with the cc genotype have white plumage in ducks. At the C locus, the recessive white allele is homozygous (c/c), effectively masking the effect of the C allele. For the genotype C/- (C/C or C/c) at the C locus, individuals have black/spotty plumage affected by the E locus. At the E locus, the E allele is dominant to e, where individuals with the E allele have black plumage and those with the e/e genotype have spotty plumage. However, individuals with the c/c genotype at the C locus have white plumage regardless of the genotype at the E locus. The C locus may be epistatic to the E locus. This pattern of plumage color inheritance was supported by the phenotypic distribution in a duck cross test [27,59]. In our study, epistasis was detected between variants in *MITF* and *MC1R* gene, which indicated that the E locus corresponds to *MC1R* and the C locus corresponds to *MITF* and that the C locus masked the effect of the E locus.

In other animals, such as chickens, mice, cattle, and pigs, the expanded black locus E is thought to correspond to *MC1R* [15,16,28,60,61]. We investigated the different mutations leading to the respective phenotypes by GWAS on duck breeds with different plumage colors. The results showed that c.52G>A and c.376G>A of *MC1R* were associated with black plumage color. We hypothesize that these two mutations on *MC1R* are relevant to the E alleles, in agreement with the results reported by YU et al. [33]. Furthermore, YU et al. also found that c.52G>A was more associated with the E allele, but this difference was not shown in our study. In our study, very few individuals of WD ducks with the black plumage phenotype showed the GG genotype at c.52G>A and c.376G>A of *MC1R*. Meanwhile, mutations of c.52G>A and c.376G>A in *MC1R* are prevalent in PT. A possible reason for this phenomenon is the presence of individuals with the spotty-feather phenotype in the Wendeng black duck while the plumage color in PT is all black. In addition, further experimental protocols and studies are needed to assess the genetic mechanisms that influence plumage color in WD.

## 5. Conclusions

We identified *MITF* and *MC1R* as candidate genes for duck plumage color. c.52G>A and c.376G>A in *MC1R* were closely related to duck black plumage. The *MITF* locus functioned upstream of *MC1R* in the determination of the white, black, and spotty colors in ducks.

## Figures and Tables

**Figure 1 genes-14-00856-f001:**
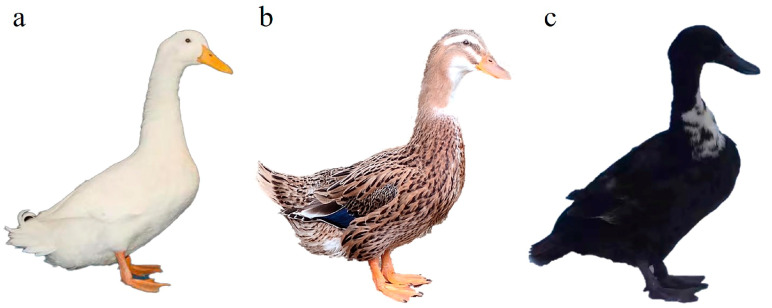
Representative images of color in duck plumage. (**a**) Duck with white plumage (Peking duck). (**b**) Duck with spotty plumage (Shanma duck). (**c**) Duck with black plumage (Wendeng black duck).

**Figure 2 genes-14-00856-f002:**
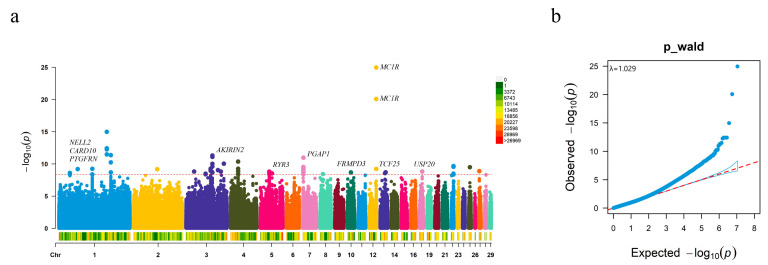
Genome-wide association study results for black plumage. (**a**) Manhattan plot of GWAS results. The *Y*-axis shows −log10 (*p*-values). The red dashed horizontal line indicates the genome-wide significance threshold (4.583099 × 10^−9^). (**b**) Q-Q plot. Lambda (λ) is the genomic expansion factor. A total of 16 ducks showed black plumage, and 56 controls were spotty.

**Figure 3 genes-14-00856-f003:**
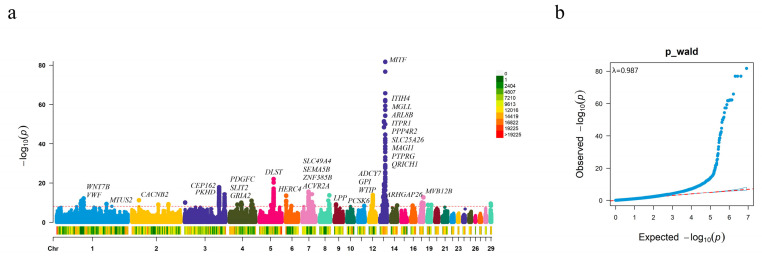
Genome-wide association study results for white plumage. (**a**) Manhattan plot of GWAS results. The *Y*-axis shows −log10 (*p*-values). The two red dashed horizontal line indicates genome- wide significance thresholds (6.042978 × 10^−9^). (**b**) Q-Q plot. Lambda (λ) is the genomic expansion factor. A total of 24 ducks showed white plumage while 56 controls were spotty.

**Table 1 genes-14-00856-t001:** Overview of duck populations used in this study.

Population	Geographic Origin	Classification	Number
Spot-billed duck (SB)	Ningxia, China	Spotty plumage	7
Mallard (MD)	Zhejiang, China	Spotty plumage	8
Peking duck (PK)	Beijing, China	White plumage	8
Maple Leaf duck (FY)	Beijing, China	White plumage	8
Cherry Valley duck (CV)	Beijing, China	White plumage	8
Wendeng black duck (WD)	Shandong, China	Black plumage	8
Putian black duck (PT)	Fujian, China	Black plumage	8
Gaoyou duck (GY)	Jiangsu, China	Spotty plumage	8
Jinding duck (JD)	Fujian, China	Spotty plumage	8
Mei duck (MEI)	Anhui, China	Spotty plumage	9
Shanma duck (SM)	Fujian, China	Spotty plumage	8
Shaoxing duck (SX)	Jiangsu, China	Spotty plumage	8
Total			96

**Table 2 genes-14-00856-t002:** Genetic polymorphisms in *MC1R* and *MITF*.

			*MITF*						*MC1R*								
Phenotype	Breeds	6.6-kb Intron Insertion	chr13:15411658 A>G			chr13:15412570 T>C			chr13:15412592 C>G			c.52 G>A			c.376 G>A		
			AA	AG	GG	TT	TC	CC	CC	CG	GG	GG	GA	AA	GG	GA	AA
Recessive white	CV	insertion	8	0	0	8	0	0	8	0	0	0	1	6	0	1	7
Recessive white	FY	insertion	8	0	0	8	0	0	8	0	0	2	3	3	1	3	4
Recessive white	PK	insertion	8	0	0	8	0	0	8	0	0	6	0	2	5	2	1
Wild-type	GY	⁄	0	0	8	0	0	8	0	0	6	8	0	0	8	0	0
Wild-type	JD	⁄	0	0	8	0	0	8	0	0	8	8	0	0	8	0	0
Wild-type	MD	⁄	0	0	8	0	0	8	0	0	8	8	0	0	8	0	0
Wild-type	MEI	⁄	0	0	9	0	0	9	0	0	9	9	0	0	9	0	0
Wild-type	SM	⁄	0	0	8	0	0	8	0	0	8	8	0	0	8	0	0
Wild-type	SX	⁄	0	0	8	0	0	8	0	0	8	8	0	0	8	0	0
Wild-type	SB	⁄	0	0	7	0	0	7	0	0	7	7	0	0	7	0	0
Extended black	PT	⁄	0	1	7	0	1	7	0	1	7	0	3	5	0	3	5
Extended black	WD	⁄	0	3	5	0	3	4	0	3	5	2	6	0	1	6	1

**Table 3 genes-14-00856-t003:** Epistatic effects of *MITF* and *MC1R*.

Group	Chr1	Gene1	SNP1	Chr2	Gene2	SNP2	*p*
	13	*MITF*	13:15411658	12	*MC1R*	12:20222793	0.226
	13	*MITF*	13:15411658	12	*MC1R*	12:20223117	0.164
white vs. non-white	13	*MITF*	13:15412570	12	*MC1R*	12:20222793	0.271
	13	*MITF*	13:15412570	12	*MC1R*	12:20223117	0.212
	13	*MITF*	13:15412592	12	*MC1R*	12:20222793	0.298
	13	*MITF*	13:15412592	12	*MC1R*	12:20223117	0.234
	13	*MITF*	13:15411658	12	*MC1R*	12:20222793	0.451
	13	*MITF*	13:15411658	12	*MC1R*	12:20223117	0.509
white vs. black	13	*MITF*	13:15412570	12	*MC1R*	12:20222793	0.438
	13	*MITF*	13:15412570	12	*MC1R*	12:20223117	0.490
	13	*MITF*	13:15412592	12	*MC1R*	12:20222793	0.451
	13	*MITF*	13:15412592	12	*MC1R*	12:20223117	0.509
	13	*MITF*	13:15411658	12	*MC1R*	12:20222793	0.025
	13	*MITF*	13:15411658	12	*MC1R*	12:20223117	0.036
white vs. spotty	13	*MITF*	13:15412570	12	*MC1R*	12:20222793	0.038
	13	*MITF*	13:15412570	12	*MC1R*	12:20223117	0.054
	13	*MITF*	13:15412592	12	*MC1R*	12:20222793	0.041
	13	*MITF*	13:15412592	12	*MC1R*	12:20223117	0.057

## Data Availability

The genome data for WD and PT ducks used in this study are available on request from the corresponding author. The remaining genome data can be found in our previous research with accession ID PRJNA686828 and PRJNA419832 [26,62].

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
