# Peer review of "Genome-Wide Association Study Reveals the Genetic Basis of Duck Plumage Colors"

_genes, 2023, doi:10.3390/genes14040856_

Round 1
Reviewer 1 Report
The study contains important data. Entry is sufficient. The method and material are sufficient. However, the explanations on the supplementary file should be made. Results discussion and conclusion are sufficient and appropriate. Suggestions and explanations given in the supplementary file should be made.

Author Response
Thank you very much for your suggestion. We have made changes to the manuscript based on your suggestions. The images were re-uploaded to ensure that the clarity and size met the requirements of the journal. Regarding the samples, our samples were previously collected by lab personnel, so there are differences. We are sure that this does not affect our results. In addition, we checked and revised the grammar of the manuscript.
The content of the supplementary document is mainly the annotated results of notable SNPS found through GWAS. We confirm that we have uploaded the supplementary file.
Author Response
Thank you very much for your suggestions. We have overhauled the manuscript based on your suggestions. Please see the attached word for the specific responses and revisions.

Reviewer 3 Report
some minor revisions required, check attachment!

Author Response
Thank you very much for your suggestions, and also for the positive comments on the results section. We have made changes according to your marks in the manuscript.
Regarding the geographical properties of the samples, we have provided the location of the sample collection in table 1. It should be noted that our samples were previously collected by laboratory personnel. In addition, our previous study also mentioned the collection of samples (Zhang et al., 2018).